# Cardiovascular disease risk factors and markers of oxidative stress and DNA damage in leprosy patients in Southern Nigeria

Iya Eze Bassey[1]*, Inyeneobong Ernest Inyang[1], Uwem Okon Akpan[1], Idongesit Kokoabasi Paul Isong[1], Bassey Edward Icha[2], Victoria Micheal Ayawan[3], Racheal Ekanem Peter[4], Hopefaith Adode Itita[1], Prince Ukam Odumusor[1], Eyoanwan Graziani Ekanem[5], Okon Ekwerre Essien[6]

1 Department of Medical Laboratory Science, Faculty of Allied Medical Sciences, College of Medical Sciences, University of Calabar, Calabar, Cross River State, Nigeria, 2 Department of Chemical Pathology, University of Calabar Teaching Hospital, Calabar, Cross River State, Nigeria, 3 Department of Microbiology, Faculty of Natural Science, Caritas University, Amorji Nike, Enugu State, Nigeria, 4 Department of Public Health, Faculty of Allied Medical Sciences, College of Medical Sciences, University of Calabar, Calabar, Cross River State, Nigeria, 5 Department of Microbiology, University of Calabar Teaching Hospital, Calabar, Cross River State, Nigeria, 6 Department of Internal Medicine, Faculty of Medicine, University of Calabar, Calabar, Cross River State, Nigeria

* iyantui@yahoo.com

**Data Availability Statement:** All relevant data are within the manuscript and its Supporting Information files.

## Abstract

Leprosy reduces quality of life of affected persons. Oxidative stress caused by reactive oxygen species may play a vital role in the pathogenesis of leprosy. This study evaluated anthropometric indices, fasting plasma glucose (FPG), lipid profile, total antioxidant capacity (TAC), total plasma peroxide (TPP), oxidative stress index (OSI), malondialdehyde (MDA), glutathione (GSH) and 8-hydroxy-2-deoxyguanosine (8-OHdg) in leprosy patients. Sixty test participants of both genders, aged 18–65years and diagnosed of multibacillary leprosy and 30 apparently healthy controls were consecutively recruited for this study. The test participants comprised of 30 patients on multidrug therapy (MDT) and 30 patients relieved from therapy (RFT). Body mass index (BMI), Waist-hip ratio (WHR), FPG, lipid profile, TAC, TPP, OSI, MDA, GSH and 8-OHdg were determined using appropriate methods. Data were analyzed using Analysis of variance; p<0.05 was considered statistically significant. The MDT group had significantly lower BMI (p = 0.0001), Total cholesterol (p = 0.001), HDL-C (p = 0.019), LDL-C (p = 0.005), TAC (p = 0.0001) and higher TPP (p = 0.001), MDA (p = 0.0001), OSI (p = 0.005) and 8-OHdg (p = 0.035) compared to the controls. The RFT group had significantly lower BMI (p = 0.001) Total cholesterol (0.0001), HDL-C (p = 0.006) LDL-C (p = 0.0001), TAC (p = 0.001) and higher WHR (p = 0.010), VLDL-C (p = 0.035), TG (p = 0.023) Atherogenic index of plasma (p = 0.0001) and TPP (p = 0.001), MDA (p = 0.0001) compared to the control group. GSH levels correlated negatively with duration of treatment (r = -0.401, p = 0.028). This study has shown that there is oxidative stress in multibacillary leprosy patients irrespective of drug treatment status. This study also shows that leprosy patients relieved from treatment may be susceptible to cardiovascular events. Antioxidants supplementation may be beneficial in the treatment of leprosy and clinical follow up on

**Funding:** The author(s) received no specific funding for this work.

**Competing interests:** The authors have declared that no competing interests exist.

patients relieved from treatment may also be necessary to monitor health status and prevent development of cardiovascular events.

## Author summary

This study shows that there are lower levels of total antioxidant capacity and higher levels of total plasma peroxide, malondialdehyde in leprosy patients undergoing multidrug therapy and those relieved from treatment and higher levels of 8-OHdg and oxidative stress index in leprosy patients undergoing multidrug therapy. This is suggestive of increased oxidative stress, in multibacillary leprosy patients irrespective of drug treatment status and increased oxidative DNA damage in those undergoing multidrug therapy. Antioxidants supplementation may be beneficial in the treatment of leprosy to protect against the effects of oxidative stress and DNA damage. Leprosy patients relieved from treatment may be susceptible to cardiovascular events as shown by higher levels of VLDL-cholesterol, triglycerides and atherogenic index of plasma observed in that group compared to controls. It therefore points to the need to monitor cardiovascular comorbidities in patients on multidrug therapy and those released from therapy.

## Introduction

Leprosy or Hansen's disease is a chronic debilitating disease caused by *Mycobacterium leprae* that affects the skin, peripheral nerves, mucosa of the upper respiratory tract, and the eyes causing progressive and permanent disabilities if not treated [1]. Although Leprosy was eliminated as a public health problem (i.e. a disease burden of < 1 case per 10 000 persons) globally in the year 2000, leprosy still remains a global public health issue [2]. Each year about 200–300 thousand people are diagnosed of leprosy and about 2–3 million people are disabled because of it [3]. Official figures from 150 countries from 6 WHO regions show that globally 7, 607, 837 persons are infected with leprosy [4]. A global registered prevalence of 192, 713 cases was reported in 2017, an increase by 20, 765 cases over that in 2016, and 210, 671 new cases were detected in the same year [5].

In Nigeria a registered prevalence of 3,234 cases was reported in 2015 and 2,892 new cases were detected in the same year [6]. With this, Nigeria ranked third among African countries with the highest burden of the disease [7]. However in 2017, prevalence of 11, 230 cases was reported and 2,447 new cases detected, with a total of 195, 875 persons infected with leprosy in the country; making Nigeria first among African countries with the highest burden of the disease in 2017 [5]. Nigeria Center for Disease Control (NCDC) moreover maintains that over 3,500 people are diagnosed with leprosy yearly in the country with about 25% of victims having some degree of disability [8].

The pathogenesis of this neurodegenerative disease is not fully understood [9]. These leprosy-associated disabilities potentially caused by damage to the peripheral nerves have been linked to a number factors including oxidative stress [10]. Excessive production of reactive oxygen species (ROS) is the major precursor of Oxidative Stress (OS). Oxidative Stress is a condition associated with an increased rate of metabolic impairment and cellular damage occurring due to derangement in the balance between ROS and the body's antioxidant system [11]. It is suggested that the possible reason for decreased antioxidant status in leprosy cases may be increased production of ROS via cell mediated immunity, as well as the free radical

producing ability of drugs used in multidrug therapy (MDT) of leprosy [12]. Of the various mechanisms that influence the pathogenesis of leprosy, oxidative stress may well be important; however, only a few studies have been done to assess the status of oxidative stress in leprosy patients via markers of oxidative stress such as Glutathione (GSH), Superoxide dismutase (SOD), and lipid peroxidase product, Malondialdehyde (MDA). 8-hydroxy-2deoxy guanosine (8-OHdG) is a DNA nucleoside which is an oxidative derivative of guanosine [10]. Measurement of the levels of 8-OHdG is important as a biomarker of oxidative stress causing cellular DNA damage. Increased levels of 8-OHdG is associated with a lot of pathologic conditions like cancer, diabetes, and hypertension [13].

Cardiovascular diseases (CVD) are the leading cause of death and disability worldwide. Together they resulted in 17.9 million deaths in 2015 up from 12.3 million in 1990 and represented 31% of all global deaths in 2016 [4]. Growing evidence indicates that chronic and acute over production of reactive oxygen species under pathophysiologic conditions is integral in the development of cardiovascular diseases [14]. Majority of cardiovascular disease results from complications of atherosclerosis, and an important initiating event for atherosclerosis may well be the oxidation of low-density lipoprotein (LDL) and the transport of oxidized low-density lipoprotein (Ox-LDL) across the endothelium into the artery wall [15]. Dyslipidaemia, hyperglycaemia, and insufficient physical activity are cardiovascular risk factors that have been associated with leprosy [16, 17].

In spite of the global interest in CVD and the growing interest in oxidative stress and its role in the aetiology of many diseases, very few studies on these have been conducted among people affected by leprosy. The few studies evaluating oxidative status in leprosy patients were conducted in India and Brazil [11, 12, 16, 18] and none amongst African people affected with leprosy. There is also dearth of information regarding the link between leprosy and the risk of cardiovascular disease in Africans, hence the need for this study. The purpose of this study was to assess the cardiovasculardisease risk factors and markers of oxidative stress and DNA damage in a native black population of leprosy patients to see if there are any deleterious changes on these markers especially in those undergoing multiple drug therapy and those relieved from treatment. The study assessed anthropometric indices, fasting plasma glucose, lipid profile, total antioxidant capacity (TAC), total plasma peroxide (TPP), oxidative stress index (OSI), malondialdehyde (MDA), glutathione (GSH) and 8-hydroxy-2-deoxyguanosine (8-OHdg) in patients with leprosy.

## Methodology

### Ethics statement

This study was carried out in accordance with the World Medical Association's Declaration of Helsinki [19]. Ethical clearance was obtained from Health Research and Ethics Committee of the Akwa Ibom State Ministry of Health, Nigeria (Ref No.: MH/PRS/99/vol.IV/452). The study participants were informed of the nature of the research and written informed consent obtained as approved by the ethics committee before they were enrolled for the study.

### Study design and subject selection

A case-control study design was used for the study and was carried out in Ekpene Ebom, Etinan LGA, in Akwa Ibom State Nigeria. A total of 90 (male and female) participants were consecutively recruited for the study. They consisted of 60 known multibacillary leprosy patients as test subjects and 30 apparently healthy individuals who did not have leprosy as controls. They were aged between 18 to 65 years and were all Nigerians. The leprosy patients were recruited from outpatients and inpatients of the Leprosy Referral Hospital in Ekpene Ebom,

and 30 apparently healthy individuals were recruited as control participants from Akwa Ibom State. The 60 test participants comprised of 30 patients on multidrug therapy (MDT) and 30 patients who have been relieved from treatment (RFT). Treatment modality for the patients was the administration of Dapsone 100 mg /daily, Rifampicin 600 mg/daily and Clofazimine 50 mg /daily for 24 months. The study was carried out from 1 September 2018 to 1 August 2019. A standard questionnaire was administered and sociodemographic data obtained.

## Measurement of anthropometric indices and blood pressure and definition of cut-off

Each participant's weight (in kilograms) and height (in metres) were measured. A weighing balance was utilized to measure weight in kilograms and a stadiometer was used to measure height in meters. Body mass index (BMI) was calculated by as the ratio of the weight to the square of height ($kg/m^2$). Normal range for BMI is 18–25 $kg/m^2$. Obesity was defined as BMI $\geq$30 $kg/m^2$. Waist circumference (in centimeters) was obtained by taking two measurements, one after inhalation and the other after exhalation at the midpoint between the top of the iliac crest and the bottom of the rib cage using an inelastic calibrated tape. Hip circumference measured at the hips and buttocks. Waist-to-hip ratio was defined as waist girth/hip circumference. Obesity was defined as Waist/hip ratio >0.9 (men), >0.85 (women). The blood pressure was measured on the right arm with a mercury sphygmomanometer (cuff size 12.5 X 40 cm) with the patient in a seated position and after a 5-minute rest. The systolic and diastolic blood pressures were recorded. Hypertension was defined as blood pressure $\geq$140/90 mmHg or the use of antihypertensive medications [20].

## Inclusion and exclusion criteria

The test participants were recruited not withstanding impairment/disability status. Leprosy patients who had just been diagnosed and had not started any treatment regimen, those refused their consent to participate and non-infected persons with any sign or symptoms of illness at the time of the study were excluded.

## Collection of sample

Seven millilitres (7ml) whole blood was collected aseptically by venepunture using a Terumo vacutainer (Terumo, Japan) from each participant after an overnight fast. Two millilitres (2ml) was dispensed into fluoride oxalate bottles for fasting plasma glucose estimation while five millilitres (5ml) was dispensed into a plain bottle, allowed to clot at room temperature, centrifuged at 3000rpm for 5minutes and serum extracted and and stored at -20˚C.

## Estimation of analytes

**Fasting plasma glucose (FPG) and lipid profile.** Fasting plasma glucose (FPG) was estimated by the glucose oxidase method of Trinder, [21]. Total cholesterol (TC) was estimated by the method of Allain *et al*., [22] and Triglycerides by the method of Bucolo and David, [23]. Estimation of high density lipoprotein cholesterol (HDL-C) was by Precipitation and cholesterol determination method of Demacker *et al*., [24] and Allain *et al*., [22] respectively. All the kits were obtained from Randox (Antrim, United Kingdom). Low Density Lipoprotein Cholesterol (LDL-C) was calculated using Friedewald's equation: LDL-C = TC—HDL-C—VLDL-C [25] while Very Low Density Lipoprotein Cholesterol (VLDL-C) was calculated using the equation: VLDL-C (mmol) = TG/2.2 provided TG was $\leq$ 4.5mmol/l [26]. Normal ranges for the variables were as follows: FPG— 3.5—5.5mmol/L; TC: 3.0—6.5mmol/L;

HDL-C: 0.9—1.5mmol/L; LDL-C: 2.5—3.8mmol/L; VLDL-C: 0.4—0.6mmol/L; TG: 1.2mmol/L. Atherogenic Index Plasma was calculated according to the formula, log (TG/HDL-C) [27]. Subjects with AIP value of > 0.11 were considered at elevated cardiovascular risk.

**Oxidative stress markers.** Determination of glutathione (GSH) was based on the methodology described by Ellman, [28] and malondialdehyde (MDA) was by the method of Buege and Aust [29]. Total antioxidant capacity was determined by the method of Koracevic *et al.*, [30]. Total plasma peroxide (TPP) was determined using the reaction of ferrous-butylated hydroxytoluene-xylenol orange complex 'FOX2' method of Miyazawa, [31] with minor modifications as described by Harma *et al.*, [32]. The ratio of total plasma peroxide (TPP) to total antioxidant capacity (TAC) was calculated as the oxidative stress index, an indicator of the degree of oxidative stress.

$$\text{OSI}\,(\%) \; = \; [\text{TPP}(\mu mol/l\,H_2O_2) \times 100)/\,[\text{TAC}\,\mu mol/l].$$

**Marker of oxidative DNA damage.** Estimation of 8-OHdG was done using Agisera's ELISA kit no: AS152887 (Agrisera, Vännäs, Sweden). The Agrisera's 8-OHdG ELISA is a competitive assay for the quantitative measurement of 8-OHdG. It utilizes an 8-OHdG coated plate and an HRP-conjugated antibody for detection which allows for an assay range of 0.94-60ng/L, with a sensitivity of 0.59ng/L. The 8-OHdG antibody used in this assay recognizes both the free 8-OHdG and the DNA incorporated 8-OHdG. All the analytes were estimated in the Chemical Pathology Laboratory of the University of Calabar Teaching Hospital, Calabar, Nigeria.

## Statistical analysis

Data was analyzed using the PAWstatistic 18, a statistical package from SPSS Inc, (Chicago IL, United States of America) and R version 4.0.0 from The R Foundation for Statistical Computing Platform, Vienna, Austria. Results were presented as mean ± standard deviation. Chi-squared test of independence was to test significant relationship between categorical variables. The assessment of the normality of the data was done using the Shapiro-Wilk's test. Comparisons of groups were made using Students' t-test and analysis of variance (ANOVA) and post hoc analysis using Least significant difference. Pearson's correlation analysis was also done. The level of significance was set at 95% confidence interval, where p-value less than 0.05 (p<0.05) was considered as statistically significant. Graphs were created with Microsoft excel 2007 version (Microsoft Corporation, US).

## Results

Table 1 shows the sociodemographic data of leprosy patients on MDT, those RFT and controls. The groups have similar sex distribution, but those relieved from treatment are older than both the controls and those undergoing treatment. The occupation of the leprosy patients was mostly farming and petty trading. However, a large proportion (60%) of those relieved from treatment were unemployed. In the MDT group, a significantly large proportion (83.3%), compiled strictly to treatment regimen with only 16.7% (n = 5) lagging in compliance. There was no co-infection with HIV in the test group. The leprosy patients relived from treatment had a significantly higher (p = 0.0001) frequency of physical deformity as well as history of leprae reactions.

Table 2 shows the comparison of anthropometric indices, blood pressures, lipid profile, atherogenic index of plasma and oxidative stress markers in leprosy patients undergoing multiple drug therapy, leprosy patients relieved from treatment and controls. The result shows that

**Table 1. Sociodemographic data of leprosy patients undergoing multidrug therapy, leprosy patients relieved from treatment and control.**

| Variable | Leprosy patients undergoing multidrug therapy n = 30 | Leprosy patients relieved from treatment n = 30 | Controls n = 30 | P-value |
|---|---|---|---|---|
| Age | 38.4± 16.80 | 49.8± 9.76[#] | 38.0± 12.79 | 0.001 |
| Occupation | Students 3 (10%) Traders 11(36.7%) Farmers 16 (53.3%) | Farmers 6(20%) Traders 6(20%) Unemployed 18(60%) | Students 7 (23.3%) Farmers 5(16.7%) Traders 5(16.7%) Civil servants 13 (43.3%) | |
| Gender | Male 19 (63.3%) Female 11 (36.7%) | Male 22 (73.3%) Female 8 (26.7%) | Male 17 (56.7%) Female 13 (43.3%) | 0.398 |
| Compliance with treatment | Compliant: 25 (85%)* Not Complaint: 5 (15%) | | | |
| Physical deformity | 4 (13.3%) | 20(66.7%) | Nil | 0.0001 |
| History of Leprae Reactions | 3(10%) | 21(70%). | Nil | 0.0001 |
| Duration of MDT (months) | 7.3±6.11 (Min (0.5); Max(23) | | | |
| Duration relieved from treatment (yr) | | 3.5±1.96 (Min (1.0); Max(8.0) | | |

there was a significant variation in the levels of BMI (p = 0.0001), WHR (p = 0.033), DBP (p = 0.048), TC (p = 0.0001), HDL-C (p = 0.013), LDL-C (p = 0.001), VLDL-C (p = 0.047), TG (p = 0.037), AIP (p = 0.001), TAC (p = 0.0001), TPP (p = 0.001), OSI (p = 0.018), MDA (p = 0.0001), GSH (p = 0.036) and 8-OHdG (p = 0.015) among the groups. There was no

**Table 2. Anthropometric indices, blood pressures, lipid profile, atherogenic index of plasma and oxidative stress markers in leprosy patients undergoing multiple drug therapy (MDT), leprosy patients relieved from treatment (RFT) and controls.**

| Parameter | Leprosy Patients | | Controls n = 30 | p-value |
|---|---|---|---|---|
| | MDT n = 30 | RFT n = 30 | | |
| BMI (kg/m$^2$) | 19.8 ± 4.30 | 20.3 ± 3.94 | 23.6± 2.58 | 0.0001* |
| Waist/Hip ratio | 0.86 ±0.04 | 0.89 ±0.08 | 0.85 ±0.05 | 0.033* |
| Systolic BP (mmHg) | 113.3 ±14.16 | 122.7 ±22.73 | 116.8 ±6.58 | 0.078 |
| Diastolic BP (mmHg) | 71.8 ±7.93 | 74.0 ±11.91 | 78.2 ±9.74 | 0.048* |
| FPG (mmol/L) | 6.19 ± 7.25 | 4.38 ± 0.64 | 4.42 ± 0.47 | 0.173 |
| Total cholesterol (mmol/L) | 3.92 ± 0.86 | 3.46 ± 0.97 | 4.68 ± 0.71 | 0.0001* |
| HDL-cholesterol (mmol/L) | 1.41 ± 0.39 | 1.37 ± 0.35 | 1.65 ± 0.41 | 0.013* |
| LDL-cholesterol (mmol/L) | 2.05 ± 0.69 | 1.59 ± 0.92 | 2.61 ± 0.58 | 0.0001* |
| VLDL-cholesterol (mmol/L) | 0.45 ± 0.25 | 0.56 ± 0.20 | 0.45 ± 0.14 | 0.047* |
| Triglycerides (mmol/L) | 0.99 ± 0.54 | 1.24 ± 0.43 | 0.98 ± 0.30 | 0.037* |
| Atherogenic index of plasma | -0.18 ±0.20 | -0.05 ±0.19 | -0.23 ±0.18 | 0.001* |
| TAC (mmol) | 1.20 ± 0.45 | 1.40 ± 0.50 | 1.94 ± 0.76 | 0.0001* |
| TPP (µmol/L) | 111.8 ± 11.66 | 112.5 ± 8.97 | 104.2 ± 6.55 | 0.001* |
| Oxidative Stress index | 12.0 ± 9.07 | 9.77 ±5.59 | 6.9 ±4.75 | 0.018* |
| MDA (mmol/L) | 0.74 ± 0.18 | 0.87 ± 0.23 | 0.52 ± 0.10 | 0.0001* |
| GSH (mg/dL) | 79.1 ± 20.34 | 79.2 ± 20.67 | 90.6 ± 29.70 | 0.111 |
| 8-OHdg (ng/L) | 9.2 ± 7.98 | 5.67 ± 1.36 | 6.44 ± 1.80 | 0.015* |

Results expressed as Mean ± SD

*significant at p<0.05

significant variation (p>0.05) in the SBP and FPG levels among the groups. Due to the fact that the RFT were older than both controls and MDT group we adjusted for age by adding age as a covariate into the analysis and the results did not change.

A post hoc analysis showed that the mean values of BMI (p = 0.0001), DBP (p = 0.016), TC (p = 0.001), HDL-C (p = 0.019) and LDL-C (p = 0.005) and TAC (p = 0.0001) were significantly lower while the mean values of TPP (p = 0.001), OSI (p = 0.005), MDA (p = 0.0001) and 8-OHdg (p = 0.035) were significantly higher in the MDT group when compared to controls. The mean values of BMI (p = 0.001), TC (p = 0.0001), HDL- C (p = 0.006), LDL-C (p = 0.0001) and TAC (p = 0.001) were significantly lower in the RFT group while the mean values of WHR (p = 0.010), VLDL-C (p = 0.035), TG (p = 0.023), AIP (p = 0.0001), TPP (p = 0.001) and MDA (p = 0.0001) were significantly higher in the RFT group when compared to the controls. Further comparison between the MDT and RFT group showed that TC (p = 0.041), LDL-C(p = 0.019) and 8-OHdg (0.006) levels were significantly lower while VLDL-C (p = 0.050), TG(p = 0.030), AIP (p = 0.011), MDA (p = 0.004) and were significantly higher in the RFT group when compared to the MDT group (Table 3).

Table 4 shows the correlation of duration of treatment and some other indices in leprosy patients undergoing multidrug therapy. Results showed a significant positive correlation between duration of treatment and TC (r = 0.611, p = 0.0001), HDL-C (r = 0.364, p = 0.048), LDL-C (r = 0.416, p = 0.022) and TG (r = 0.364, p = 0.048). However, a negative significant correlation was observed between duration of treatment and GSH (r = -0.401, p = 0.028).

## Discussion

The study assessed fasting plasma glucose, lipid profile, total antioxidant capacity, total plasma peroxide, oxidative stress index, malondialdehyde, glutathione and 8-OHdg in patients with leprosy on mutltidrug therapy (MDT) and those relieved from therapy (RFT). The WHO health Organization has assiduously worked towards a leprosy-free world through its Global Leprosy Strategy 2016–2020 [33] and so has the National Tuberculosis and Leprosy Control Programme in Nigeria. However, more has to be done towards reintegrating those who have been cured of the disease into society. From our study, 60% of the leprosy patients that were relieved from treatment were unemployed. More than two thirds of the patients were males who should supposedly be breadwinners for their families.

Significantly lower levels of total cholesterol (TC), HDL-cholesterol (HDL-C) and LDL-cholesterol (LDL-C) were observed in patients on MDT and those RFT compared to the controls and significantly higher levels of VLDL-cholesterol (VLDL-C), triglycerides (TG) and Atherogenic index of plasma (AIP) in patients RFT compared to the controls. The low TC levels obtained in this study agrees with the findings of Sheikh et al., [34] and Nega et al., [35] who also detected lower TC levels in multibacillary form of the disease. However Silva et al., [16] reported normal levels of serum TC in both paucibacilliary and multibacilliary forms of the disease and other authors have even reported an increase in serum TC in patients with lepromatous leprosy [36]. The low serum TC observed in this study may be due to increase utilization of cholesterol by *M. leprae* for nutrition and persistence of the bacilli [18] or due to hepatic involvement as evidenced by several studies indicating hepatic damage by *M. leprae* bacilli [36]. It is also suggested that in lepromatous leprosy, free fatty acids are increased due to microbial modulation of lipoprotein lipase activity, which exerts an inhibitory effect on hepatic lipogenesis directly and indirectly on cholesterol synthesis by inhibiting pyruvate dehydrogenase enzymes of the liver [37].

The leprosy patients in both groups (MDT and RFT) had significantly lower HDL-C levels compared to the controls. This result differs from the findings of Silva et al., [16] and Sheikh

**Table 3. Comparison of anthropometric indices, blood pressures, lipid profile, atherogenic index of plasma and oxidative stress markers in leprosy patients undergoing multiple drug therapy (MDT), leprosy patients relieved from treatment (RFT) and controls using post hoc analysis.**

| Parameter | Groups | | Mean Difference | Std Error | p-value |
|---|---|---|---|---|---|
| | MDT n = 30 | Controls n = 30 | | | |
| BMI (kg/m$^2$) | 19.8 ± 4.30 | 23.6± 2.58 | **-3.800** | **0.951** | **0.0001** |
| Diastolic BP (mmHg) | 71.8 ±7.93 | 78.2 ±9.74 | -6.367 | 2.581 | 0.016 |
| Total cholesterol (mmol/L) | 3.92 ± 0.86 | 4.68 ± 0.71 | -0.760 | 0.221 | 0.001 |
| HDL-cholesterol (mmol/L) | 1.41 ± 0.39 | 1.65 ± 0.41 | -0.237 | 0.099 | 0.019 |
| LDL-cholesterol (mmol/L) | 2.05 ± 0.69 | 2.61 ± 0.58 | -0.560 | 0.192 | 0.005 |
| TAC (mmol) | 1.20 ± 0.45 | 1.94 ± 0.76 | 0.733 | 0.151 | 0.0001 |
| TPP (μmol/L) | 111.8 ± 11.66 | 104.2 ± 6.55 | 7.643 | 2.401 | 0.001 |
| Oxidative Stress index | 12.0 ± 9.07 | 6.9 ±4.75 | 5.047 | 1.739 | 0.005 |
| MDA (mmol/L) | 0.74 ± 0.18 | 0.52 ± 0.10 | 0.210 | 0.046 | 0.0001 |
| 8-OHdg (ng/L) | 9.2 ± 7.98 | 6.44 ± 1.80 | 2.772 | 1.290 | 0.035 |
| | RFT n = 30 | Controls n = 30 | | | |
| BMI (kg/m$^2$) | 20.3 ± 3.94 | 23.6± 2.58 | **-3.343** | **0.951** | 0.001 |
| Waist/Hip ratio | 0.89 ±0.08 | 0.85 ±0.05 | 0.041 | 0.015 | 0.010 |
| Total cholesterol (mmol/L) | 3.46 ± 0.97 | 4.68 ± 0.71 | -1.217 | 0.221 | 0.0001 |
| HDL-cholesterol (mmol/L) | 1.37 ± 0.35 | 1.65 ± 0.41 | -0.277 | 0.099 | 0.006 |
| LDL-cholesterol (mmol/L) | 1.59 ± 0.92 | 2.61 ± 0.58 | -1.020 | 0.192 | 0.0001 |
| VLDL-cholesterol (mmol/L) | 0.56 ± 0.20 | 0.45 ± 0.14 | 0.111 | 0.052 | 0.035 |
| Triglycerides (mmol/L) | 1.24 ± 0.43 | 0.98 ± 0.30 | 0.260 | 0.112 | 0.023 |
| Atherogenic Index of plasma | -0.05 ±0.19 | -0.23 ±0.18 | 0.182 | 0.049 | 0.0001 |
| TAC (mmol) | 1.40 ± 0.50 | 1.94 ± 0.76 | -0.540 | 0.151 | 0.001 |
| TPP (μmol/L) | 112.5 ± 8.97 | 104.2 ± 6.55 | 8.330 | 2.401 | 0.001 |
| MDA (mmol/L) | 0.87 ± 0.23 | 0.52 ± 0.10 | 0.347 | 0.046 | 0.0001 |
| | MDT n = 30 | RFT n = 30 | | | |
| Total cholesterol (mmol/L) | 3.92 ± 0.86 | 3.46 ± 0.97 | -0.457 | 0.221 | 0.041 |
| LDL-cholesterol (mmol/L) | 2.05 ± 0.69 | 1.59 ± 0.92 | 0.460 | 0.192 | 0.019 |
| VLDL-cholesterol (mmol/L) | 0.45 ± 0.25 | 0.56 ± 0.20 | -0.102 | 0.052 | 0.050 |
| Triglycerides (mmol/L) | 0.99 ± 0.54 | 1.24 ± 0.43 | -0.247 | 0.112 | 0.030 |
| Atherogenic Index of plasma | -0.18 ±0.20 | -0.05 ±0.19 | -0.128 | 0.049 | 0.011 |
| MDA (mmol/L) | 0.74 ± 0.18 | 0.87 ± 0.23 | -0.137 | 0.046 | 0.004 |
| 8-OHdg (ng/L) | 9.2 ± 7.98 | 5.67 ± 1.36 | 3.550 | 1.256 | 0.006 |

*et al.* [34] who reported normal and significantly higher serum HDL-C levels in leprosy patients respectively but corroborate the findings of Gupta *et al.*, [37] and Nega *et al.*, [35] who detected significantly lower HDL-C levels in patients with the lepromatous form of the disease compared to the controls. It is suggested that this low serum HDL-C level observed could be due to increase in the levels of cytokines such as tumor necrosis factor (TNF-α) which lowers the HDL levels [35, 37]. Pro- inflammatory immune response against microbial infections up regulates the production of cytokines like TNF -α, IL-1, IL-2 & IL-6 as a host protective mechanism against bacterial assault [38]. TNF-α is a critical host-protective cytokine against mycobacterial diseases that decreases HDL levels [39] and studies have shown that TNF-α levels are increased in PB lesions and episodes of reverse reaction [40,41]. The precise mechanism by which TNF-α decrease HDL levels is uncertain and is likely to involve multiple mechanisms. It

**Table 4. Correlation of duration of treatment and some other indices in leprosy patients undergoing multiple drug therapy.**

| Parameter | Index | r-value | p-value |
|---|---|---|---|
| Duration of treatment (months) | Body mass index | 0.272 | 0.146 |
| | Systolic BP | 0.152 | 0.422 |
| | Diastolic BP | -0.297 | 0.111 |
| | Fasting plasma glucose | -0.077 | 0.687 |
| | Total cholesterol | 0.611 | 0.0001* |
| | HDL-cholesterol | 0.364 | 0.048* |
| | LDL-cholesterol | 0.416 | 0.022* |
| | VLDL-cholesterol | 0.342 | 0.064 |
| | Triglycerides | 0.364 | 0.048* |
| | Atherogenic Index of plasma | 0.166 | 0.379 |
| | Total antioxidant capacity | 0.230 | 0.221 |
| | Total plasma peroxide | -0.331 | 0.074 |
| | Oxidative Stress index | -0.260 | 0.165 |
| | Malondialdehyde | -0.272 | 0.147 |
| | Glutathione | -0.401 | 0.028* |
| | 8-OHdg | 0.148 | 0.148 |

*Significant at p<0.05

decreases the production of apo-A1, enhances HDL degradation by macrophages, decreases Lecithin Cholesterol Acyltransferase (LCAT) activity, causes an increase in serum amyloid A production, induces secretory phospholipase A2 (sPLA2) and endothelial cell lipase (HDL-metabolizing enzymes) which may all result in decreased HDL levels [38, 42].

Significantly lower levels of LDL-C were observed in multibacillary leprosy patients irrespective of drug therapy status when compared to the controls in this study. This agrees with the findings of a study Sheikh *et al.*, [34]. Normal LDL-C levels were however observed by Silva *et al.* [16] in MB leprosy patients when compared to controls. In the present study, serum LDL-C level was also shown to be significantly lower in participants relieved from therapy when compared to those on MDT. The low LDL-C levels observed in this study may be due to impact of infection on hepatic lipogenesis. Nwosu & Nwosu [36] observed significantly high levels of alanine amino transferase in MB leprosy and Dhavalshankh *et al.*, [43] observed significantly lower levels of albumin in lepromatous leprosy and these studies concluded that leprosy is associated with liver damage. The liver is the principal organ involved in the lipid metabolism and has a major role in the control of plasma LDL-C level since most LDL-C receptors are present in the liver; therefore its invasion by leprae bacilli may alter lipid metabolism [44].

This study showed significantly higher levels of VLDL and TG in MB leprosy patients relieved from treatment when compared to controls. This is similar to the work of Nega *et al.*, [35]. However, normal triglyceride levels in leprosy patients with no significant variation in VLDL levels have been reported by other authors [16, 45]. The higher levels of triglyceride and VLDL-C cholesterol may be attributed to several mechanisms, including reduction of TG hydrolysis, lipopolysaccharide- and pro-inflammatory cytokine-induced de novo free fatty acid production and TG synthesis in the liver as well as reduction of lipoprotein lipase activity thus resulting in reduced VLDL clearance and increased TG levels [38]. The decreased clearance of TG-rich lipoproteins may be due to continued induction of cell-mediated inflammatory response caused by the presence of *M. leprae* antigens even after completion of therapy

[46]. Some cytokines (TNF-α, IL-1, IL-2, IL-6) have been shown to decrease the synthesis of lipoprotein lipase, the key enzyme that metabolizes triglycerides in circulation, and inflammatory responses increases the production of these cytokines. Studies have also shown that inflammation also increases angiopoietin like protein 4, an inhibitor of lipoprotein lipase activity, which would further block the metabolism of VLDL and TG [47]. It may also be important to note that the increase in may also be due to age–related factors since those relieved from treatment were older than the controls, however this effect is mild as the statistical differences observed still remained after adjusting for age.

This work also showed positive significant correlation in TC, HDL-C and LDL-C levels with duration of treatment. This agrees with the observation that MDT may benefit hepatic lipogenesis and plasma lipid and lipoprotein abnormalities/levels may even return towards normal following recovery from the infection [42]. This study also showed significant increase in TG levels following increase in treatment duration. Some anti-inflammatory drugs have effects on lipid metabolism that are independent of the reduction in inflammation. For example, high dose glucocorticoid treatment results in an increase in serum triglyceride and VLDL levels due to the increased production and secretion of VLDL by the liver [42]. Because all MDT regimens for leprosy contain Clofazimine, which has strong anti-inflammatory activity [48], MDT may therefore increase VLDL secretion in the liver leading to increase in plasma TG levels.

Atherogenic index of plasma is a novel indicator of dyslipidemia and associated cardiovascular disease [49]. This study showed a significantly higher AIP in multibacillary leprosy patients when compared to controls. Inflammation and infections induce a variety of alterations in lipid metabolism that if prolonged could contribute to the increased risk of atherosclerosis. The most common changes are decreases in serum HDL-C and increases in triglyceride levels. In addition to affecting serum lipid levels, inflammation also adversely affects lipoprotein function. LDL is more easily oxidized as the ability of HDL to prevent the oxidation of LDL is diminished leading to atherosclerosis [42]. AIP was also significantly higher in the RFT group when compared to patients on MDT. This agrees with recent findings that cytokines mediated inflammatory responses, perhaps to persistent *M*. *leprae* antigen, still occur in leprosy patients even after being declared cured [35, 46]. Inflammation reduces serum HDL-C and increases TG levels and could contribute to the increased risk of atherosclerosis [42].

Results from this study also showed that the BMI of the both groups of leprosy patients compared to the controls was significantly lower indicating wasting, maybe due to malnutrition [12]. However, the WHR of leprosy patients relieved from treatment was significantly higher when compared to the control group. This agrees with the work of Silva *et al*., [16] who observed a higher waist circumference in multibacillary leprosy patients compared to controls. This suggests that WHR may be a better indicator of adiposity in these patients and leprosy.

The observed higher levels of TG, VLDL-C, AIP, WHR and lower levels of HDL-C observed in patients RFT compared to controls, suggests that leprosy patients who are relieved from treatment may be predisposed to developing a cardiovascular disease. This may result in a double burden of disease as well as increased mortality for those who suffer from this disease.

In this study oxidative stress markers assessed included total antioxidant capacity (TAC), total plasma peroxide (TPP), oxidative stress index (OSI), malondialdehyde (MDA), glutathione (GSH) while 8-hydroxy-2-deoxyguanosine (8-OHdg) was assessed as a marker of oxidative DNA marker. The significantly lower levels of TAC and GSH and higher levels of TPP, MDA in both groups of patients (MDT and RFT) and higher OSI and 8-OHdg in only the MDT group when compared to the controls suggests the presence of increased oxidative stress. These findings agree with the works of Prabhakar *et al*., [11] who showed that there is

oxidative stress in MB leprosy cases, irrespective of their treatment status. Oxidative stress has been suggested to play a vital role in the pathogenesis of leprosy [12]. The leprosy patients exhibit increased oxidative stress which could mediate inflammatory episodes, depressed cell-mediated immune response, organ damage as well as degeneration of nerves in these patients [12]. Oxidative stress not only occurs in newly diagnosed cases but also in patients who have finished the therapy [50]. Higher MDA levels observed in RFT group suggests that there may be continued induction of ROS by persistent *M. leprae* antigen or remnant bacilli even after completion of therapy [46]. Significantly higher MDA levels indicate that increased lipid per-oxidation and DNA damage due to free radical mediated injury occurs in leprosy patients [51]. A major defence against microbial infection is the macrophage system [52]. The macro-phages infected by mycobacteria show increased phagocytosis, enzyme activity and oxygen consumption known as "respiratory burst," [12]. Microbial killing by macrophages is associated with a burst of respiratory activity that also leads to production of ROS [51]. Another probable factor that may encourage lipid peroxidation in leprosy patients may the oxidative stress that is associated with amyloid formation and deposition [52]. Amyloid deposits have been reported in some leprosy patients with prevalence from autopsy studies ranging from 3.9% and 31% [53]. Major targets of ROS-associated peroxidation are the polyunsaturated fatty acids (PUFA) found in membrane lipids. PUFA is degraded by free radicals to form mal-ondialdehyde (MDA) therefore MDA in serum serves as a marker of cellular damage due to free radicals [12]] and high serum MDA levels is indicative of increased lipid peroxidation and DNA damage by free radicals [54].

Antioxidant status is an index of oxidative stress. High level of oxidative stress is also shown by the low cellular antioxidant status [50] and decrease in antioxidant defense may be one of the reasons for increased levels of ROS and subsequent tissue damage in lepromatous spec-trum [55]. In this study, both the RFT and MDT leprosy patients had lower TAC levels com-pared to controls although there was no significant variation in their GSH levels. Antioxidants which play a central role in maintaining the cells redox state both enzymatically and non-enzy-matically by neutralizing free radicals, notably superoxide radicals, hydroxyl radicals, nitric oxide and carbon [56]. Studies have also shown that GSH is considered a potent inhibitor of lipid peroxidation process, and thus regulates the MDA content [57]. The comparable GSH levels in the leprosy patients compared with the controls contradicts findings in studies by Prabhakar *et al.*, [11] and Osadalor and Okosun, [58] who all reported lower GSH levels in lep-rosy patients compared to controls. The reason for this is not clear.

Interestingly, there was significant negative correlation between GSH levels and duration of treatment. Decrease in GSH levels were observed with increase in treatment duration showing induction of oxidative stress with increased duration of MDT and only the MDT group had higher OSI and 8-OHdg compared to the controls. This may be due to the free radical produc-ing effect of Dapsone, one of drugs in the MDT regimen [58]. Schlacher *et al.*, [59] showed by their study that hydroxylation of Dapsone by hepatic cytochromes generate the metabolites (N-hydroxylamine DDS-NHOH and monoacetyl-hydroxylamine MADDS-NHOH) that pro-duces free radicals by electron transfer. It may also have been utilized for replenishment of other crucial antioxidants such as vitamin E and C, which would get oxidized during the course of their antioxidant action [57]. They also suggested that MDT can thus reduce the activity of some antioxidant enzyme causing ROS accumulation and thereby exhausting the production of adaptive oxidants defenses, resulting in a higher oxidative stress index, lipid per-oxidation as well as higher 8-OHdg levels in this group. The higher 8-OHdg suggests a predis-position of those undergoing treatment to oxidative DNA damage.

Our findings suggest that the increased oxidative stress observed may therefore be due to the body's response to active infection; depletion of antioxidants defenses following therapy in

the MDT and perhaps continued induction of free radicals by *M. leprae* antigens even after completion of therapy [46]. The increased OS observed in these patients can mediate inflammatory episodes, organ damage, depressed cell mediated immune response and degeneration of nerves in leprosy patients [12].

A limitation of this study was that we did not estimate the *M. leprae* bacterial load as this may have helped in shedding more light on the results obtained from those relieved from treatment i.e. to ascertain if there was persistence of infection. In addition, those relieved from treatment were significantly older than both the controls and those undergoing treatment. Though we tried to adjust for age, it may affect the interpretation of our result as age is a risk factor for cardiovascular disease and oxidative stress. Another limitation was the fact that we had no treatment–naïve group because this was not allowed by the hospital's protocol because of the highly infectious nature of the disease. But it would have helped us observe the pattern of changes in the assessed analytes before, during and after treatment.

## Conclusion

This study shows that there are lower levels of total antioxidant capacity and higher levels of total plasma peroxide, malondialdehyde in leprosy patients undergoing multidrug therapy and those relieved from treatment and higher levels of 8-OHdg and oxidative stress index in leprosy patients undergoing multidrug therapy. This is indicative of increased oxidative stress, in multibacillary leprosy patients irrespective of drug treatment status and increased oxidative DNA damage in those undergoing treatment. Leprosy patients relieved from treatment may be susceptible to cardiovascular events as shown by higher levels of VLDL-cholesterol, triglycerides and atherogenic index of plasma observed in that group compared to controls. Antioxidants supplementation may be beneficial in the treatment of leprosy to protect against the effects of oxidative stress and clinical follow up on patients relieved from treatment may also be necessary to monitor health status and prevent development of cardiovascular events.

## Supporting information

**S1 Data.**
(XLSX)

## Acknowledgments

We would like to thank all the participants who took part in this study, the staff of the leprosy referral hospital, Ekpene Obom for giving us access to their patients and Department of Chemical Pathology Laboratory, University of Calabar Teaching Hospital, Calabar for the use of their facilities.

## Author Contributions

**Conceptualization:** Iya Eze Bassey, Idongesit Kokoabasi Paul Isong, Okon Ekwerre Essien.

**Data curation:** Inyeneobong Ernest Inyang, Victoria Micheal Ayawan, Racheal Ekanem Peter.

**Formal analysis:** Iya Eze Bassey, Inyeneobong Ernest Inyang.

**Funding acquisition:** Iya Eze Bassey, Inyeneobong Ernest Inyang.

**Investigation:** Inyeneobong Ernest Inyang, Bassey Edward Icha, Hopefaith Adode Itita, Prince Ukam Odumusor.

**Methodology:** Iya Eze Bassey, Bassey Edward Icha.

**Project administration:** Inyeneobong Ernest Inyang, Uwem Okon Akpan.

**Resources:** Uwem Okon Akpan.

**Software:** Iya Eze Bassey, Uwem Okon Akpan.

**Supervision:** Iya Eze Bassey, Idongesit Kokoabasi Paul Isong.

**Validation:** Iya Eze Bassey.

**Writing – original draft:** Iya Eze Bassey, Inyeneobong Ernest Inyang, Uwem Okon Akpan, Idongesit Kokoabasi Paul Isong, Victoria Micheal Ayawan, Racheal Ekanem Peter, Hope-faith Adode Itita, Prince Ukam Odumusor.

**Writing – review & editing:** Iya Eze Bassey, Uwem Okon Akpan, Idongesit Kokoabasi Paul Isong, Bassey Edward Icha, Victoria Micheal Ayawan, Racheal Ekanem Peter, Eyoanwan Graziani Ekanem, Okon Ekwerre Essien.

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
