## [Decision Letter · Decision Letter 0]

23 Jun 2020

Dear Dr. Bassey,

Thank you very much for submitting your manuscript "Cardiovascular Risk Factors and Markers of Oxidative Stress and DNA Damage in Leprosy Patients in Southern Nigeria." for consideration at PLOS Neglected Tropical Diseases. As with all papers reviewed by the journal, your manuscript was reviewed by members of the editorial board and by several independent reviewers. In light of the reviews (below this email), we would like to invite the resubmission of a significantly-revised version that takes into account the reviewers' comments. 

We cannot make any decision about publication until we have seen the revised manuscript and your response to the reviewers' comments. Your revised manuscript is also likely to be sent to reviewers for further evaluation.

Sincerely,

Alberto Novaes Ramos Jr, M.D., M.P.H., Ph.D.

Guest Editor

Mathieu Picardeau

Deputy Editor

Reviewer's Responses to Questions

**Key Review Criteria Required for Acceptance?**

**Methods**

-Are the objectives of the study clearly articulated with a clear testable hypothesis stated?

-Is the study design appropriate to address the stated objectives?

-Is the population clearly described and appropriate for the hypothesis being tested?

-Is the sample size sufficient to ensure adequate power to address the hypothesis being tested?

-Were correct statistical analysis used to support conclusions?

-Are there concerns about ethical or regulatory requirements being met?

Reviewer #1: To strengthen the work, it is necessary to align the analyzes carried out with a well-defined hypothesis and objective. I suggest that the authors include the purpose of the study in the introduction. In lines 102-105 the authors specify details about the collection of information and anthropometric data, however, they do not mention the measurement of blood pressure and the calculation of the waist-to-hip ratio. I suggest including this information as well as the reference values associated with it. The absence of reference values for the evaluated markers made it difficult to read and analyze the tables, so I suggest the authors to detail this information. Exclusion criteria topics can be added to the inclusion criteria topic. Lines 108-112 repeat some of the information mentioned in the previous topic (Study design and subject selection). I suggest that the authors review the need for repetition of information. In the statistical analysis, the authors do not inform whether they performed the assessment of the normality of the data and which test was used. This information must be clear since it supports the choice for parametric tests. I also suggest that the authors specify which regression analysis was used.

Reviewer #2: The objectives and hypothesis are clearly stated and the study design is appropriate to address the objective. Overall, the population is clearly described. No sample size calculation was given, but there seems to be a sufficient number of participants in each group (n = 30). The statistical analysis support the conclusions, however it is not clear whether the groups have similar characteristics (e.g. age, sex distribution). Ethical permission has been received and this was reported.

Reviewer #3: Well defined and nicely described methodology section of the study

**Results**

-Does the analysis presented match the analysis plan?

-Are the results clearly and completely presented?

-Are the figures (Tables, Images) of sufficient quality for clarity?

Reviewer #1: The results follow the analysis plan. I recommend the authors briefly explore the differences between groups (MDT, RFT, and controls) for the results presented in Tables 1 and 2.

Table 1, which presents the sociodemographic data of the study subjects, did not include information on the age variable. The inclusion of this variable is relevant, as it is an important factor in the analysis of cardiovascular risk and oxidative stress.

The analysis of the waist-to-hip ratio without stratifying according to gender makes difficult to interpret since it has different reference values between men and women.

Reviewer #2: The analysis match the analysis plan. The results are presented, but there is some information missing in table 1 (e.g. compliance) and the scale/value of some variables is not clear (e.g. duration of treatment in days/weeks/months?). The quality of the tables is sufficiently.

Reviewer #3: Nicely described results section as well as nicely presented analysis of results.

**Conclusions**

-Are the conclusions supported by the data presented?

-Are the limitations of analysis clearly described?

-Do the authors discuss how these data can be helpful to advance our understanding of the topic under study?

-Is public health relevance addressed?

Reviewer #1: The discussion presented is interesting and states the limitations and generalizations of the study.

In lines 263-269, the authors discuss the role of pro-inflammatory cytokines in decreasing HDL levels. Since the patients included had multibacillary leprosy, the literature refers that the immune response elicited produces predominantly cytokines from the Th2 pattern. Studies have shown that TNF-α levels are increased in PB lesions and episodes of reverse reaction. These information should be taken into consideration when elaborating on the topic.

Reviewer #2: The conclusions are supported by the data presented and are clearly stated. The limitations of the analysis are clearly described and the authors show a great understanding of the topic. The public health relevance is addressed and recommendations are made for interventions.

Reviewer #3: Conclude the study section very good and given a key message for leprosy control program.

**Editorial and Data Presentation Modifications?**

Reviewer #1: Abstract

Remove the abbreviation ROS, as the word is mentioned only once in the summary;

The abbreviations BMI, TC, WHR, and AIP have no mention of the original word.

Introduction

Line 45-46: specify the period to which the total number of people infected with leprosy refers to the sentence;

Line 52: present the terms referred to in the abbreviation ROS.

Methodology

Line 126: add the abbreviation FPG to the term fasting plasma glucose.

Results

Line 182: mean BPD values were significantly lower in the MDT group when compared to the control group;

Line 185: correct the p-value for the TAC.

Reviewer #2: Minor revisions are needed for the methods, results and discussion.

I recommend to make the following revisions: 

Methods: 

Please describe what materials and methods you used. For example in line 118 'whole blood was collected' how did you do this and with what type of material from what company? 

Also for methods that are described elsewhere like line 126 'method of Trinder' did you use the exact same material from the exact same company and distribution center?

What do you mean with minor modifications in line 141? 

Please provid details on your protocols. For example in line 122 'frozen until assayed' at what temperature, -80 or -20? 

Specify the ELISA kit you used and from what company, to do this correctly you need to give the companies full name, the city/state the company is located and the country the company is located and put this within brackets behind the material. So for example: Agisera's ELISA kit no:12345 (Agisera, Vännäs, Sweden). 

This also applies to the software you use SPSS in line 155 and Microsoft in line 160

Results

Table 1: Did you test for differences between the groups, for example in gender distribution? And what about the age of the patients? It seems like there might be a difference in the gender distribution, did you check whether this affected your results and did you correct for this accordingly in your results? 

Table 1: Not necessary to put disease form and HIV in it, since you selected for MB patients and for no other ilnesses, right? 

Table 1: I miss the compliance in the table, this is described in line 166. 

Table 1: how did you assess physical deformities, on what scale? This is also not described in the methods. 

Table 2: Are the Calf F and Crit. F really needed? 

Table 4: The duration of treatment is in days/weeks/moths/years? 

Discussion: 

Line 250 : 'both forms' what forms is meant by this. Be careful with the different 'forms' of leprosy, leprametous and MB are different terms since they belong to different classification systems but these terms overlap. 

Line 365 - 368: What might be causing these different results?

Reviewer #3: Not required

**Summary and General Comments**

Reviewer #1: It is a relevant study that contributes to the improvement of care for patients with leprosy by bringing evidence about the increase in cardiovascular risk and oxidative stress in MB patients in Nigeria. It adds knowledge about the theme and points to the need to monitor cardiovascular comorbidities in patients on MDT and released from therapy.

Reviewer #2: Overall, a very interesting manuscript! The introduction and discussion are well-written and show great understanding of the topic. This is very valuable since it is the first study in Africa evaluating oxidative status in leprosy patients, while leprosy is still prevalent in parts of this continent. The results are very interesting and relevant and the study has been performed according to ethical guidelines.

Reviewer #3: The manuscript Number PNTD-D-20-00597, entitled "Cardiovascular Risk Factors and Markers of Oxidative Stress and DNA Damage in Leprosy Patients in Southern Nigeria”. This is an interesting and exhaustive article on the cardiovascular risk factors in leprosy patients. The article is relatively well written. However, there are various fallacies in this article. 

I would like the authors to consider the following points to make before this article can be processed further.

Comments on the manuscript PLOSNTD- major revision

01. How was the Multibacillary (MB) leprosy cases diagnosed?

02. The treatment for MB leprosy cases should be described in detail, including dosage and duration.

03. How was the cardiac evaluation done?

04. Was ECG and Echocardiography as well as X-ray chest done for evaluation of cardiac status?

05. What could have been the exact cause of oxidative stress and DNA damage?

06. Amyloidosis is an important cause of cardiac, renal and hepatic deterioration in MB leprosy patients. So mention has to be made of this entity in the discussion section.

07. In the introduction section, Epidemiology of the disease should be briefly described, including the number of cases worldwide as well as in Nigeria.

08. Mention has to be made about the leprosy elimination program in the world as well as in Africa and Nigeria in the discussion section.

09. The English and grammatical errors need to be corrected.

All the above mentioned points need to be taken care of before further processing.

PLOS authors have the option to publish the peer review history of their article (what does this mean?). If published, this will include your full peer review and any attached files.

Reviewer #1: No

Reviewer #2: No

Reviewer #3: No
---

## [Decision Letter · Decision Letter 1]

1 Aug 2020

Dear Dr. Bassey,

Thank you very much for submitting your manuscript "Cardiovascular disease Risk Factors and Markers of Oxidative Stress and DNA Damage in Leprosy Patients in Southern Nigeria." for consideration at PLOS Neglected Tropical Diseases. As with all papers reviewed by the journal, your manuscript was reviewed by members of the editorial board and by several independent reviewers. The reviewers appreciated the attention to an important topic. Based on the reviews, we are likely to accept this manuscript for publication, providing that you modify the manuscript according to the review recommendations. 

Sincerely,

Alberto Novaes Ramos Jr, M.D., M.P.H., Ph.D.

Guest Editor

Mathieu Picardeau

Deputy Editor

Reviewer's Responses to Questions

**Key Review Criteria Required for Acceptance?**

**Methods**

-Are the objectives of the study clearly articulated with a clear testable hypothesis stated?

-Is the study design appropriate to address the stated objectives?

-Is the population clearly described and appropriate for the hypothesis being tested?

-Is the sample size sufficient to ensure adequate power to address the hypothesis being tested?

-Were correct statistical analysis used to support conclusions?

-Are there concerns about ethical or regulatory requirements being met?

Reviewer #1: Yes.

Reviewer #2: The methods are much more clear now that details have been added on the procedures and materials.

Reviewer #3: Described methodology of the study is well defined.

**Results**

-Does the analysis presented match the analysis plan?

-Are the results clearly and completely presented?

-Are the figures (Tables, Images) of sufficient quality for clarity?

Reviewer #1: Yes.

Reviewer #2: As expected, the socio-demographic characteristics of the groups significantly differ. These characteristics can have major impact on the outcome parameters measured in this study. The increased aged in the group that is relieved from therapy can for example affect the results significantly. If I understand it correctly, these characteristics have not been taking along in the analysis of the outcome parameters, while they should be in my opinion. In addition, the paragraph describing table 2 and 3 (line 207-226) is really hard to read because of all the p-values in the text. Finally the last parameter in table 2 'duration relieved from therapy' is probably displaced, because now it is placed under the group that still receives therapy.

Reviewer #3: Results section is nicely presented.

**Conclusions**

-Are the conclusions supported by the data presented?

-Are the limitations of analysis clearly described?

-Do the authors discuss how these data can be helpful to advance our understanding of the topic under study?

-Is public health relevance addressed?

Reviewer #1: Yes.

Reviewer #2: Only few limitations have been described, while this study has multiple. Especially the fact that the characteristics of the groups significantly differ and that this might affect the results is important to mention in the discussion. Please elaborate the limitations in the discussion.

Reviewer #3: The conclusion section of the study is excellent and forward a key message for leprosy control program.

**Editorial and Data Presentation Modifications?**

Reviewer #1: The manuscript needs formatting, in some sentences, there is an excess of periods (lines 201, 300), excessive use of spaces (lines 175,133, 136, etc.) and a different color in the text (269-273).

Reviewer #2: Please correct for the differences between the groups a priori (e.g. age, deformities) during your analysis of the outcome parameters. There is clearly a link between several cardiovasculair risk factors/markers of oxidative stress and age. So if the groups differ so significantly (p = 0.001) you can not 'just' compare the outcome parameters of these groups as you did in table 3. Please also discuss this in the discussion within the limitation section.

Reviewer #3: Not required

**Summary and General Comments**

Reviewer #1: The authors did a great job of reviewing and endeavored to improve the manuscript. The reviewers' suggestions were taken into account, and the doubts raised were clarified.

Reviewer #2: No comments

Reviewer #3: The manuscript Number PNTD-D-20-00597, entitled "Cardiovascular Risk Factors and Markers of Oxidative Stress and DNA Damage in Leprosy Patients in Southern Nigeria”. The revised article is relatively well written.

PLOS authors have the option to publish the peer review history of their article (what does this mean?). If published, this will include your full peer review and any attached files.

Reviewer #1: No

Reviewer #2: No

Reviewer #3: No
---

## [Decision Letter · Decision Letter 2]

26 Aug 2020

Dear Dr. Bassey,

We are pleased to inform you that your manuscript 'Cardiovascular disease Risk Factors and Markers of Oxidative Stress and DNA Damage in Leprosy Patients in Southern Nigeria.' has been provisionally accepted for publication in PLOS Neglected Tropical Diseases.

Best regards,

Alberto Novaes Ramos Jr, M.D., M.P.H., Ph.D.

Guest Editor

Mathieu Picardeau

Deputy Editor

Reviewer's Responses to Questions

**Key Review Criteria Required for Acceptance?**

**Methods**

-Are the objectives of the study clearly articulated with a clear testable hypothesis stated?

-Is the study design appropriate to address the stated objectives?

-Is the population clearly described and appropriate for the hypothesis being tested?

-Is the sample size sufficient to ensure adequate power to address the hypothesis being tested?

-Were correct statistical analysis used to support conclusions?

-Are there concerns about ethical or regulatory requirements being met?

Reviewer #1: Since the authors included a new analysis adjusted for age, it is necessary to give more details in the methodology section.

Reviewer #2: Methods are clear now

Reviewer #3: Method section of the study are well defined and nicely described.

**Results**

-Does the analysis presented match the analysis plan?

-Are the results clearly and completely presented?

-Are the figures (Tables, Images) of sufficient quality for clarity?

Reviewer #1: The other reviewers pointed out that the age variable could be a possible confounding factor in the study. The authors then performed a new analysis adjusted for age. Therefore, there is now a need to include the results of this new analysis in a supplementary file or even throughout the text.

Reviewer #2: Results are correct now

Reviewer #3: Results section of the study is nicely presented.

**Conclusions**

-Are the conclusions supported by the data presented?

-Are the limitations of analysis clearly described?

-Do the authors discuss how these data can be helpful to advance our understanding of the topic under study?

-Is public health relevance addressed?

Reviewer #1: Yes.

Reviewer #2: Limitations have been described properly now

Reviewer #3: Conclude of the study is section very good and given a key message for leprosy control program

**Editorial and Data Presentation Modifications?**

Reviewer #1: line 452 - change subtitle formatting

line 348 - where it reads "It may also be important to note that the increase in may (...) "- a word seems to be missing from the sentence.

Reviewer #2: No

Reviewer #3: Not needed

**Summary and General Comments**

Reviewer #1: No comments.

Reviewer #2: I recommend to accept this manuscript

Reviewer #3: The manuscript Number PNTD-D-20-00597, entitled "Cardiovascular Risk Factors and Markers of Oxidative Stress and DNA Damage in Leprosy Patients in Southern Nigeria”. This is an interesting article and relatively well presented.

PLOS authors have the option to publish the peer review history of their article (what does this mean?). If published, this will include your full peer review and any attached files.

Reviewer #1: No

Reviewer #2: No

Reviewer #3: No

---

## [Editor Report · Acceptance letter]

2 Oct 2020

Dear Dr. Bassey,

We are delighted to inform you that your manuscript, "Cardiovascular disease Risk Factors and Markers of Oxidative Stress and DNA Damage in Leprosy Patients in Southern Nigeria.," has been formally accepted for publication in PLOS Neglected Tropical Diseases.

Best regards,

Shaden Kamhawi

co-Editor-in-Chief

Paul Brindley

co-Editor-in-Chief
